# Immunohistochemistry Screening of Different Tyrosine Kinase Receptors in Canine Solid Tumors—Part I: Proposal of a Receptor Panel to Predict Therapies

**DOI:** 10.3390/ijms25158438

**Published:** 2024-08-02

**Authors:** Denner Santos Dos Anjos, Patrick Antônio Sonaglio Civa, Juliana Werner, Igor Simões Tiagua Vicente, Carlos Eduardo Fonseca-Alves

**Affiliations:** 1Department of Veterinary Surgery and Animal Reproduction, Universidade Estadual Paulista (UNESP), Botucatu 18618-681, Brazil; denner.anjosoncology@gmail.com; 2Medicalvet Veterinary Clinic, Chapecó 89802-272, Brazil; patrickapple2@hotmail.com; 3Werner and Werner Laboratory, Curitiba 80540-160, Brazil; juliana@werner.vet.br; 4VetPrecision Laboratory, Botucatu 18608-970, Brazil; igor.tiagua@gmail.com; 5Institute of Veterinary Oncology, IOVET, São Paulo 05027020, Brazil

**Keywords:** dogs, epidermal growth factor, immunohistochemistry, platelet-derived growth factor receptor, receptor tyrosine kinase, target therapy, vascular endothelial growth factor

## Abstract

The use of tyrosine kinase inhibitors (TKI) has been growing in veterinary oncology and in the past few years several TKI have been tested in dogs. However, different from human medicine, we lack strategies to select patients to be treated with each TKI. Therefore, this study aimed to screen different tumor subtypes regarding TKI target immunoexpression as a predictor strategy to personalize the canine cancer treatment. It included 18 prostatic carcinomas, 36 soft tissue sarcomas, 20 mammary gland tumors, 6 urothelial bladder carcinomas, and 7 tumors from the endocrine system. A total of 87 patients with paraffin blocks were used to perform immunohistochemistry (IHC) of human epidermal growth factor receptor 2 (HER-2), epidermal growth factor receptors 1 (EGFR1), vascular endothelial growth factor receptor 2 (VEGFR-2), platelet derived growth factor receptor beta (PDGFR-β), c-KIT, and extracellular signal-regulated kinase 1/2 (ERK1/ERK2). The immunohistochemical screening revealed a heterogeneous protein expression among histological types with mesenchymal tumors showing the lowest expression level and carcinomas the highest expression. We have demonstrated by IHC screening that HER2, EGFR1, VEGFR-2, PDGFR-β and ERK1/ERK2 are commonly overexpressed in dogs with different carcinomas, and KIT expression is considered relatively low in the analyzed samples.

## 1. Introduction

In the last decade, the use of tyrosine kinase inhibitors (TKI) has been growing in veterinary oncology and the search for different techniques with predictive potential is the focus of different studies. The identification of cellular surface receptors can identify different cellular pathways in cancers from both humans and dogs [1]. In humans, overexpression of surface receptors such as epidermal growth factor receptors 1 and 2 (EGFR and ERBB2/HER2) was associated with the development and progression of certain types of cancer, including breast cancer (BC), head and neck cancer (HNSCC), stomach, pulmonary adenocarcinoma, and bladder cancer [2,3,4,5,6,7]. In human BC, immunohistochemistry (IHC) is the first screening test to classify tumors as human epidermal growth factor receptor 2 (HER2) overexpressing and direct to trastuzumab treatment [8]. Recently, HER2 expression and ERRB2 amplification were assessed in a group of aggressive salivary duct carcinomas and the HER2 score was considered promising to direct HER2-specfic therapies [9].

In veterinary medicine, EGFR and ERBB2 have been reported in dogs and cats with cancer [10]. In cats, HER2 overexpression was previously observed in feline mammary carcinoma (FMC), oral squamous cell carcinoma (FOSCC), pulmonary carcinoma, transitional cell carcinoma, hepatocellular carcinoma, and nasal carcinoma [8,9,10,11,12,13]. Its increased expression has been reported to be correlated with tumorigenicity in FMC as well as in canine mammary tumors being associated with decreased disease-free interval and overall survival time [10,11,12]. In addition, the main ligand of EGFR, epidermal growth factor (EGF), was evaluated in canine mammary carcinoma cell lines to play a role in the proliferation, migration, angiogenesis, and survival [13], suggesting that the EGFR pathway may be involved in the stimulation of angiogenesis as observed by Carvalho et al. [14] in which expression of EGFR was correlated with microvessel density.

A recent veterinary study investigated the HER2 expression and ERBB2 amplification in canine mammary gland tumors and found HER2 overexpression correlated with ERBB2 amplification [15]. The investigation of HER2 immunoexpression leads to several translational opportunities. Previously, our research group has investigated the role of lapatinib on HER2-positive and negative mammary gland tumor cell lines [16]. We found a low lapatinib IC_50_ for cell lines with HER-positive expression and also a negative correlation between *HER2* gene expression and lapatinib antitumor response. Therefore, cell lines with higher *HER2* gene expression had a lower lapatinib IC_50_ [16]. After several studies, lapatinib has been proposed as a drug therapy for dogs with carcinomas showing HER-2 overexpression, including urothelial [17] and pulmonary carcinomas [18]. 

Based on this previous data, HER2 overexpression seems to direct therapy with lapatinib. Thus, Maeda et al. [17] conducted a clinical trial on the clinical use of lapatinib in canine patients with urothelial carcinomas (UC) according to its HER2 expression. The authors found an excellent antitumor response, progression-free disease, and overall survival based on HER2 expression. Even patients with HER2 overexpression without ERBB2 amplification were associated with the best antitumor response. Therefore, HER2 overexpression independent of ERBB2 amplification predicated lapatinib antitumor response [17].

The first step in selecting a marker for a predictive panel is screening this marker in several different tumor types, assessing its immunolocalization (expression pattern) and percentage of positive cells. Thus, this research aimed to screen different tumor types for human epidermal growth factor receptor 2 (HER-2), epidermal growth factor receptors 1 (EGFR1), vascular endothelial growth factor receptor 2 (VEGFR-2), platelet derived growth factor receptor beta (PDGFR-β), c-KIT, and extracellular signal-regulated kinase 1/2 (ERK1/ERK2) immunoexpression to propose a panel of antibodies predictive for the use of TKI drugs. 

## 2. Results

HER-2, EGFR1, VEGFR-2, PDGFR-β, c-KIT, and MAPK (ERK1/ERK2) were selected as potential markers based on the TKI drug previously tested in dogs in vivo. The immunohistochemistry analysis used only validated antibodies in canine tissue and covered the targets of lapatinib (HER-2 and EGFR1), sorafenib (ERK1/ERK2, VEGFR-2, and PDGFR-β), toceranib (VEGFR-2, PDGFR-β, and c-KIT) and vemurafenib (ERK1/ERK2). Then, a panel called the Multikinase Panel^®^ (Painel Multiquinase^®^ in Portuguese) was registered at the Brazilian National Institute of Industrial Property (INPI). 

Each marker was assessed according to the previous literature regarding its immunolocalization. Then, the markers were scored based on the percentage of positive cells (number of positive cells/total of cells). For HER-2 expression, only membranous staining was considered; membranous and cytoplasmic expression was considered for EGFR-1, and membranous, cytoplasmic, and nuclear c-KIT expression was assessed separately. Regarding VEGFR-2 and PDGFR-β, membranous and cytoplasmic expression was considered and for MAPK (ERK1/ERK2) nuclear and cytoplasmic expression were found. Figure 1 summarizes the different scores for each marker. 

In the tumors analyzed, no expression was seen in infiltrating inflammatory or stromal cells and only expression by cancer cells was considered. Among the tumor group evaluated, 6 out of 20 mammary gland tumors with normal adjacent tissue were identified. EGFR1, VEGFR-2, PDGFR-β, c-KIT, and MAPK (ERK1/ERK2) were negative in normal tissue from the six cases. However, HER-2 expression was found in normal glands in all six cases. 

The immunohistochemical screening revealed a different protein expression for the different histological subtypes. For canine prostate cancer (PC), 17 out of 18 cases were positive for membranous HER-2 and in 13 cases, the expression was over 50% (+3 and +4). On the other hand, for EGFR1, four cases were negative, and one case showed only scattered cells for EGFR1 with membranous expression (1% of the cells). In total, 10 out of 18 cases showed EGFR1 expression over 50% (+3 and +4). All tissue samples were positive for VEGFR-2, with 9 out of 18 samples showing expression over 50%. Regarding PDGFR-β, 11 out of 18 cases were positive, with only 1 case showing over 50% expression. For c-Kit, only one case showed 10% expression (+1) and the remaining cases were negative. For ERK1/ERK2, 8 out of 18 cases showed positive expression, with only 2 cases showing over that 50% expression. The expression pattern of each case can be found in Table 1.

Soft tissue sarcomas (STSs) presented a very small number of cases with positive expression for the different markers. None of the cases showed HER2, EGFR1, and c-KIT expression; seven cases showed VEGFR2 positive expression but lower than 50%; seven cases had PDGFR-β positive expression. Three cases showed ERK1/ERK2 positive expression but lower than 50%. The complete expression pattern for each marker can be found in Table 1.

For the canine mammary gland tumors (MGT), it included comedocarcinoma, solid carcinoma, carcinoma in mixed tumors, tubular carcinoma, complex carcinoma, inflammatory carcinoma, and micropapillary carcinoma. Each corresponding tumor can be observed in Table 1. In total, 11 out of 20 cases were positive for HER2, with 9 cases showing HER2 expression over 50% (+3, and +4). For EGFR1, 13 out of 20 cases were positive, but only 2 cases presented EGFR1 expression over 50%. For VEGFR-2, 16 out of 20 cases were positive, with 10 samples showing expression over 50%. Regarding PDGFR-β, 11 out of 20 were positive with only 3 cases presenting expression more than 50%. No cases were positive for c-KIT. For ERK1/ERK2, 8 out of 20 cases showed positive expression, with 2 cases over 50% expression. The complete expression pattern is described in Table 1. 

For urothelial carcinoma (UC), all tissue samples (n = 6) were positive for HER-2, and EGFR-1, with all of them over 50% expression. Regarding VEGFR-2, five out of six were positive, with four over 50% expression. For PDGFR-β, only one sample was positive but less than 25% (1+). For c-KIT, two out of six were positive with less than 25% expression (1+). For ERK1/ERK2, five out of six were positive, with four showing over 50% expression. The complete expression pattern is described in Table 1. 

Finally, it was included seven tumors from the endocrine system. Thyroid carcinomas, insulinomas, mammary neuroendocrine carcinoma, hepatic neuroendocrine carcinoma and pancreatic solid carcinoma were evaluated. Only one case (case 7) was positive for HER2 as well as for EGFR1, but with low expression (1+). VEGFR-2 was positive for all endocrine tumors with three cases showing over 50% expression (cases 4, 6, and 7). For PDGFR-β, five out of seven were positive with two cases more than 50% expression (cases 1 and 4). All patients were negative for c-KIT expression. Finally, for ERK1/ERK2, five out of seven were positive, with two showing over 50% expression. The complete expression pattern is described in Table 1.

## 3. Discussion

In human oncology, IHC is widely used as a predictive marker for cancer treatment [19,20,21,22] assisting as a predictive marker for the use of TKI and immunotherapies, including breast cancer [23], small cell lung carcinoma [24], endometrial cancer [25] and renal cell carcinoma [26]. On the other hand, in veterinary medicine, no well-established predictive marker is used for any specific therapy [27]. In veterinary medicine, toceranib (TOC) is the most used TKI in clinical practice for several cancer subtypes. TOC has its main targets for c-KIT, VEGFR-2, and PDGFR-β. After the proposal of TOC for the treatment of canine mast cell tumors, several papers have investigated separately the expression of c-KIT, VEGFR-2, and PDGFR-β by IHC in different tumor subtypes (without clinical treatment) claiming that IHC expression could predict TOC therapy [28,29,30]. One previous study associated the expression of c-KIT, VEGFR-2, and PDGFR-β and TOC antitumor response in vitro [31] and several studies have investigated TOC antitumor response in vivo, without association with c-KIT, VEGFR-2, and PDGFR-β tumor expression [32,33,34]. Therefore, we lack studies in veterinary medicine associating TOC antitumor response with its respective targets. 

In human medicine, IHC is usually assessed using precise algorithms or well-stabilized methods for expression measurement. However, this is made based on different published studies and associating the IHC expression for each marker with the antitumor response [35,36,37]. Therefore, to establish the HER-2 IHC score and determine which scores patients could be treated with trastuzumab in BC-affected patients, several studies associating HER-2 IHC expression with antitumor response were needed [38,39,40]. In veterinary medicine, instead of creating a specific score for each tumor in dogs, based on the IHC pattern of canine tumors and the antitumor response of different markers, we just copy previously established information for humans and apply it to canine tumors. Therefore, we do not know if the scores proposed for the different tumors in humans represent the same for dogs. A major example of this paradox is the investigation of HER-2 expression in canine and feline mammary gland tumors [41,42,43]. Several other researchers have investigated HER-2 in situ hybridization and HER-2 expression trying to resemble human BC and all of them failed [15,44]. After years of research and using human scores with no success, we should change our approach. 

For this reason, we proposed here the evaluation of each IHC score based on a percentage of expression and immunolocalization as a first step, and we believe that a specific score only should be applied as a predictive marker for TKI antitumor response when the treatment was conducted, and specific statistics tests are made to create a score for each marker. In our study, we initially screened different tumor types regarding the immune expression of different tyrosine kinase receptors including a total of 87 patients. From this screening, it was possible to conclude that STS expressed very low kinase receptors precluding the use of target drugs. For this reason, STS may not be included in the pilot prospective, multicenter, randomized clinical trial. Taking this into account, all other solid tumors may be included in future pilot studies (part 2) to correlate the IHC expression and clinical response. Most of the tumors expressed HER2 and EGFR1 in our screening results including MGT, PC, and UC. The investigation of overexpression of HER2 and also ERBB2 amplification in canine mammary gland tumors have been associated with ERBB2 amplification [15]. Thus, this led to several translational opportunities. Even in patients with HER2 overexpression without ERBB2 amplification was associated with antitumor response [17]. Therefore, HER2 overexpression independent of ERBB2 amplification predicated lapatinib antitumor response. The inflammatory and stromal cells infiltrated intratumorally were usually negative. These can be related to the nature of the markers (tyrosine kinase overexpressed in epithelial or cancer cells) [23,24,25,26,27]. Thus, the expression assessed was closely related to cancer cells and not infiltrating cells. 

Regarding canine PC, the evaluation of VEGF-A and its receptor (VEGFR-2) have been proven to be an independent prognostic factor making it possible to use it for therapeutic applications [16]. In our study, we also observed that all canine PC expressed VEGFR-2 with half of them showing more than 50% of positive expression. Investigations in tissue samples of canine PC and its transcriptome profile showed higher levels of VEGF-A and VEGFR-2 in PC when compared with normal prostate. However, no correlation was observed between the gene and protein expression levels of VEGFR-2, but patients with low expression levels exhibited a higher survival time. Furthermore, another study showed in cell lines of PC that treatment with TOC decreased gene expression levels of *VEGFR-2* demonstrating the efficacy of TOC in inhibiting the *VEGFR-2* gene [45].

Although PDGFR-β was observed in 11 out of 18 cases, only 1 case showed over 50% expression. If this lower expression (<50%) would benefit the patient with target drugs still needs further confirmation. Recently, it was evaluated the effect of TOC in two canine PC cell lines, showing positive cytoplasmatic expression of PDGFR-β but with different responses to TKI. This difference could be explained by possible different driving pathways to tumor proliferation [45].

About the expression profile of UC, most of the samples were positive over 50% for HER-2, EGFR-1, VEGFR-2, and ERK1/ERK2. The same results were also observed in the literature for paraffin-embedded tissues with positive staining for VEGFR-2 [30,46,47]. However, in contrast to our results about PDGFR-β, in which we did not find a significant positivity in UC tissues (only one patient expressed positive), in the veterinary literature, a significant number of samples expressed PDGFR-β [30,48,49]. This multiple kinase receptor expression may be a potential therapeutic target for UCs such as the use of sorafenib, which targets rapidly accelerated fibrosarcoma (RAF), VEGFR-1, VEGFR-2, VEGFR-3, PDGFR-β, and KIT leading to its direct anti-tumor effect and inhibition of angiogenesis, already observed in vivo experiments in nude mice with canine UCs [47]. 

Finally, a retrospective study showed the clinical benefit of TOC in dogs with insulinoma with a response criterion for solid tumors of 66.7% (complete remission, partial response, or stable disease) [48]. However, all the studies evaluating insulinomas do not evaluate tyrosine kinase receptors [48,49,50]. This situation may occur due to expensive assessment of IHC, or the absence of macroscopic lesions in some patients precluding the pathological and molecular confirmation properly. Another point to comment about the TOC in insulinomas is that the mechanism by which TKI affects glucose homeostasis remains unclear. In these two dogs evaluated, an absence of expression was observed for HER2, EGFR-1, c-KIT, and ERK1/ERK2, with positive expression in both for VEGFR-2 and PDGFR-β. By which molecular and signalment mechanisms that TKI acts in improving clinical signs attributed to hypoglycemia in dogs that received TOC remains to be further investigated. Another important point to address is that tumors from the endocrine system showed different expression patterns for tyrosine kinase receptors with most of them positive for VEGFR-2, PDGFR-β, and ERK1/ERK2 making these receptors potential drug targets. 

## 4. Materials and Methods

### 4.1. Screening of Different Tyrosine Kinase Receptors in Canine Solid Tumors

This study was performed to screen different tumor types regarding the immunoexpression of different tyrosine kinase receptors; formalin-fixed paraffin-embedded tissue blocks from a total of 87 patients were used to perform IHC for HER-2, EGFR1, VEGFR-2, PDGFR-β, c-KIT, and ERK1/ERK2. This study was performed under the national and international guidelines for the use of animals in research and the study was approved by the Research Ethics Committee (Protocol number 0039/2020). 

### 4.2. Selection of Solid Tumor Cases

A group of solid tumors were retrieved from different clinical, surgical, and pathological specialized services. As criteria of inclusion, it was considered patients subjected to biopsy and sufficient tissue to perform IHC and only primary solid tumors were included. Metastatic lesions were not included in the study, and some patients had developed metastatic disease. In total, 18 prostatic carcinomas were retrieved from the Werner and Werner Pathology laboratory, 36 soft tissue sarcomas, and 20 mammary gland carcinomas from VetPrecision veterinary laboratory and Institute of Veterinary Oncology—IOVET. Six urothelial bladder carcinomas, and seven tumors from the endocrine system (thyroid carcinoma, insulinoma, solid pancreatic carcinoma, mammary and hepatic neuroendocrine carcinoma) were obtained from Eletro-Onkovet service. Since this study proposes an immunohistochemical predictive panel (registered as Multikinase panel^®^), the focus of this research was to screen different tumor types to identify which ones could benefit from the panel. Therefore, patients were selected who underwent tumor removal or biopsy for the diagnosis. Since the main goal was to standardize the Multikinase panel^®^, recommendations were followed by the previous veterinary literature [51] for each immunohistochemistry step.

### 4.3. Cross-Reactivity with Canine Tissue

Since we are proposing an immunohistochemistry panel focused on clinical application, we look at the cross-reactivity of each primary antibody with canine tissue. For EGFR1 and ERK1/ERK2 antibodies, the cross-reactivity was provided by the manufacturer to present a specific reaction with canine tissue. Moreover, the manufacturer also indicates directly the use of both antibodies for immunohistochemistry in paraffin tissue. The c-Kit antibody was previously validated by the literature using Western blot [52]. The PDGFR-β cross-reactivity with canine tissue was based on protein homology provided by the manufacturer and validated by Western blot [53]. The VEGFR-2 antibody was previously validated [31] and our research group has demonstrated the high homology between human and canine VEGFR-2 sequencing and applicability in paraffin tissue sections [16]. The HER-2 antibody was previously validated by Tsuboi et al. [54] using Western blot. 

### 4.4. Immunohistochemistry

The horseradish peroxidase (HRP) polymer system and 3,3′-diaminobenzidine (DAB) chromogen techniques were used for immunohistochemistry [55]. Briefly, the paraffin blocks were cut and tissue sections were extended on charged slides for immunohistochemistry (StarFrost, Knittel, Braunschweig, Germany). Tissue deparaffinization was performed using xylene and antigen retrieval was performed in a pressure chamber (Muscae Plus, Erviegas, Indaiatuba, SP, Brazil) for approximately 30 min (with pH 6.0 citrate solution). Then, the slides were subjected to endogenous peroxidase blocking with an 8% commercial solution (Peroxidase Blocking, Easypath Diagnostics, Indaiatuba, SP, Brazil). The primary antibodies were used for immunohistochemistry: c-Kit (CD117 polyclonal, catalog number: A4502, Dako Cytomation, Carpinteria, CA, USA) at a dilution of 1:100, PDGFR-β (Rabbit polyclonal, catalog number: 3162S, Cell Signaling Technology, Danvers, MA, USA) at 1:200 dilution, anti-VEGFR-2 (Rabbit polyclonal, catalog number: A5609, Santa Cruz Biotechnology, Dallas, TX, USA) at 1:150 dilution, anti-EGFR (Mouse monoclonal, catalog number: MA5-12875, ThermoFisher, Waltham, MA, USA), anti-HER2 (Rabbit polyclonal, catalog number: A0485, Dako Cytomation, Carpinteria, CA, USA) at 1:400 dilution, anti-ERK1/ERK2 (Rabbit polyclonal, catalog number: 602-330, ThermoFisher, Waltham, MA, USA) at 1:100 dilution. The Envision polymer system (Dako Cytomation, Carpinteria, CA, USA) was used as the secondary antibody. As positive controls, a mast cell tumor was used for c-Kit, a placenta for PDGFR-β, VEGFR-2, and EGFR1, a skin for HER2, and a normal testis for ERK1/ERK2. The selection of the control tissues was made on the expression profile provided by The Human Protein Atlas (https://www.proteinatlas.org, accessed on 1 July 2023). Isotype immunoglobulin was used as the negative control at the same concentration as the primary antibody.

### 4.5. Immunohistochemistry Analysis and Selection of the Tyrosine Kinase Inhibitors

The immunohistochemistry interpretation was made according to the previous veterinary literature for the establishment of a new diagnostic test [51]. Therefore, we standardized antigen immunolocalization and a semiquantitative score to estimate the percentage of positive cells. For HER2, only membranous staining was considered, while membranous and cytoplasmic expression were considered for EGFR1, VEGFR-2, and PDGFR-β. Nuclear and cytoplasmic expression were considered for MAPK (ERK1/ERK2) and membranous, cytoplasmic focal, or cytoplasmic diffuse were considered for c-KIT. Regarding expression semiquantitative analysis, immunohistochemistry was assessed based on a scoring system ranging from 0 to 4, based on each immunostaining, as follows: 0, no staining; 1+, 5–25% of positive cells; 2+, 26–50% of positive cells; 3+, 51–75% of positive cells; and 4+, >75% of positive cells. 

## 5. Conclusions

In conclusion, it was demonstrated by IHC screening that HER2, EGFR1, VEGFR-2, PDGFR-β, c-KIT, and ERK1/ERK2 revealed a different protein expression for the different histological tumors. Furthermore, it needs to be further investigated if tissue tumors from patients that expressed more than 50% for receptor targets would present better clinical response than those that do not have, suggesting a potential of predicting therapy. 

## Figures and Tables

**Figure 1 ijms-25-08438-f001:**
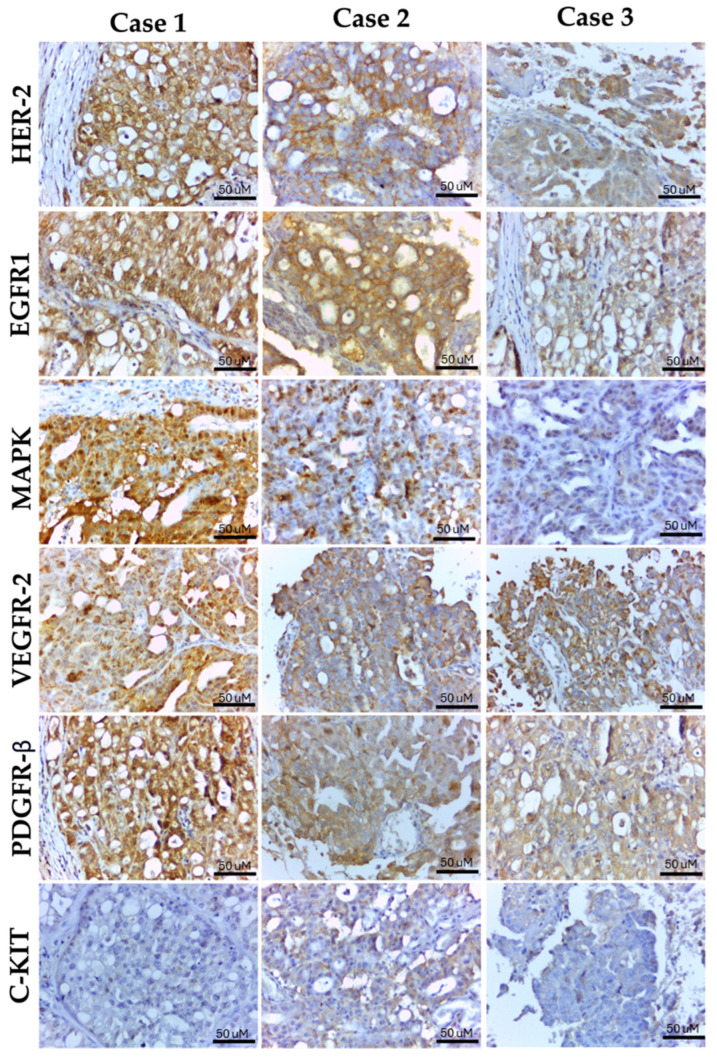
Immunoexpression of each marker in urinary bladder carcinoma (case 1), prostatic carcinoma (case 2), and carcinoma in mixed tumor (Case 3). Case 1 showed human epidermal growth factor receptor 2 (HER-2) membranous, epidermal growth factor receptors 1 (EGFR1) membranous and cytoplasmic, extracellular signal-regulated kinase 1/2 (ERK1/ERK2) cytoplasmic and nuclear expression, and both membranous and cytoplasmic for vascular endothelial growth factor receptor 2 (VEGFR-2) and platelet derived growth factor receptor beta (PDGFR-β) expression in over than 75% of neoplastic cells. This case showed a negative c-KIT expression. Case 2 had positive membranous HER-2 expression and positive nuclear and cytoplasmic ERK1/ERK2 expression in more than 50% of neoplastic cells. This case also showed positive membranous and cytoplasmic EGFR1, VEGFR-2, and PDGFR-β expression in over than 75% of neoplastic cells. Also, cytoplasmic expression was identified for c-KIT in less than 25% of neoplastic cells. Case 3 presented HER-2 membranous expression in less than 25% of neoplastic cells (there is more cytoplasmic than membranous expression and cytoplasmic expression was not assessed). This case also showed ERK1/ERK2 cytoplasmic and nuclear expression in lower than 25% of neoplastic cells. Membranous and cytoplasmic EGFR1, VEGFR-2, and PDGFR-β expression in over 75% of neoplastic cells have also been identified, and scattered cytoplasmic c-KIT expression in less than 10% of neoplastic cells was found.

**Table 1 ijms-25-08438-t001:** *Pattern of immunoexpression of different tyrosine kinase receptors from prostatic* carcinomas, soft tissue sarcomas, mammary gland carcinomas, urothelial bladder carcinomas, and endocrine/neuroendocrine tumors.

ID	HER2	EGFR-1	VEGFR-2	PDGFR-β	c-Kit	ERK1/ ERK2
Prostatic Carcinomas						
Case 1	0	0	2+	1+	0	0
Case 2	2+	0	1+	0	0	0
Case 3	2+	0	2+	0	0	0
Case 4	2+	0	2+	0	0	0
Case 5	3+	1+	3+	0	0	2+
Case 6	3+	2+	3+	2+	0	2+
Case 7	4+	3+	2+	2+	0	0
Case 8	3+	2+	2+	2+	0	0
Case 9	4+	3+	2+	1+	1+	0
Case 10	3+	3+	4+	0	0	3+
Case 11	2+	4+	3+	2+	0	1+
Case 12	3+	3+	3+	2+	0	0
Case 13	4+	3+	3+	0	0	0
Case 14	4+	3+	4+	3+	0	4+
Case 15	3+	3+	2+	2+	0	0
Case 16	4+	2+	3+	2+	0	2+
Case 17	3+	3+	4+	2+	0	1+
Case 18	3+	3+	2+	0	0	1+
Soft Tissue Sarcomas						
Case 1	0	0	0	1+	0	0
Case 2	0	0	0	0	0	0
Case 3	0	0	0	0	0	0
Case 4	0	0	1+	0	0	0
Case 5	0	0	0	0	0	0
Case 6	0	0	0	0	0	0
Case 7	0	0	0	0	0	0
Case 8	0	0	1+	1+	0	0
Case 9	0	0	0	0	0	0
Case 10	0	0	0	0	0	0
Case 11	0	0	1+	0	0	0
Case 12	0	0	0	0	0	0
Case 13	0	0	0	1+	0	0
Case 14	0	0	0	0	0	0
Case 15	0	0	2+	0	0	0
Case 16	0	0	0	0	0	0
Case 17	0	0	0	0	0	0
Case 18	0	0	1+	2+	0	0
Case 19	0	0	0	0	0	0
Case 20	0	0	0	0	0	2+
Case 21	0	0	0	0	0	0
Case 22	0	0	0	0	0	0
Case 23	0	0	0	1+	0	0
Case 24	0	0	0	0	0	0
Case 25	0	0	0	0	0	0
Case 26	0	0	1+	0	0	0
Case 27	0	0	0	0	0	0
Case 28	0	0	0	0	0	1+
Case 29	0	0	0	2+	0	0
Case 30	0	0	0	0	0	0
Case 31	0	0	0	0	0	0
Case 32	0	0	1+	0	0	0
Case 33	0	0	0	0	0	0
Case 34	0	0	0	1+	0	1+
Case 35	0	0	0	0	0	0
Case 36	0	0	0	0	0	0
Mammary Gland Carcinomas						
Case 1 (comedocarcinoma)	3+	2+	3+	2+	0	0
Case 2 (solid carcinoma)	3+	3+	3+	2+	0	0
Case 3 (carcinoma in mixed tumor)	0	1+	0	1+	0	0
Case 4 (solid carcinoma)	3+	2+	3+	0	0	2+
Case 5 (Tubular carcinoma)	0	1+	3+	0	0	0
Case 6 (carcinoma in mixed tumor)	0	0	4+	3+	0	0
Case 7 (complex carcinoma)	0	0	2+	3+	0	3+
Case 8 (inflammatory carcinoma)	0	1+	3+	1+	0	3+
Case 9 (complex carcinoma)	0	0	2+	1+	0	0
Case 10 (tubular carcinoma)	0	1+	2+	0	0	0
Case 11 (complex carcinoma)	3+	0	0	1+	0	1+
Case 12 (tubular carcinoma)	0	0	2+	0	0	0
Case 13 (micropapillary carcinoma)	3+	2+	1+	0	0	0
Case 14 (inflammatory carcinoma)	3+	2+	0	0	0	0
Case 15 (carcinoma in mixed tumor)	0	0	0	0	0	0
Case 16 (solid carcinoma)	4+	3+	4+	3+	0	2+
Case 17 (solid carcinoma)	2+	0	3+	1+	0	2+
Case 18 (solid carcinoma)	3+	2+	3+	2+	0	2+
Case 19 (solid carcinoma)	2+	2+	2+	0	0	1+
Case 20 (solid carcinoma)	3+	2+	3+	0	0	0
Urothelial Bladder Carcinomas						
Case 1	3+	3+	3+	0	1+	4+
Case 2	4+	3+	3+	0	0	3+
Case 3	3+	3+	2+	0	0	0
Case 4	4+	3+	4+	0	0	4+
Case 5	4+	3+	3+	0	1+	4+
Case 6	3+	3+	0	1+	0	2+
Endocrine/neuroendocrine Tumors						
Case 1 (thyroid carcinoma)	0	0	1+	3+	0	2+
Case 2 (thyroid carcinoma)	0	0	2+	1+	0	2+
Case 3 (insulinoma)	0	0	2+	0	0	0
Case 4 (insulinoma)	0	0	4+	2+	0	0
Case 5 (mammary neuroendocrine carcinoma)	0	0	1+	0	0	2+
Case 6 (hepatic neuroendocrine carcinoma)	0	0	4+	4+	0	4+
Case 7 (pancreatic solid carcinoma)	1+	1+	4+	1+	0	3+

## Data Availability

Data is contained within the article.

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
