# Peer review of "Immunohistochemistry Screening of Different Tyrosine Kinase Receptors in Canine Solid Tumors—Part I: Proposal of a Receptor Panel to Predict Therapies"

_ijms, 2024, doi:10.3390/ijms25158438_

Round 1

Reviewer 1 Report (Previous Reviewer 3)

Comments and Suggestions for Authors

Dear Authors, the manuscript deals with a current topic of great interest in veterinary oncology.

I invite you, first of all, to use impersonal verb forms in the text to increase the scientific depth of the research.

I point out a few points that can be supported by further information/details to implement the scientific quality of the manuscript, which I indicate below:

Line 67 at the end of the sentence, after the full stop, insert a bibliographic citation.

Figure 1 the bar indicating the magnification scale contains no micrometer indication. I invite you to refine the photo panel.

Reading the article carefully, in all the epithelial and mesenchymal tumours studied, even if malignant, I found no reference to potential neoplastic metastases. If your study sample does not include metastatic tumours, I consider it relevant to mention in section 4.1.1 Selection of solid tumour cases.

Comments on the Quality of English Language

The quality of English language needs a refining. The authors must use impersonal verbal form for improving the scientific soundness of the manuscript.

Author Response

Dear Authors, the manuscript deals with a current topic of great interest in veterinary oncology. I invite you, first of all, to use impersonal verb forms in the text to increase the scientific depth of the research. I point out a few points that can be supported by further information/details to implement the scientific quality of the manuscript, which I indicate below:

Rev: Line 67 at the end of the sentence, after the full stop, insert a bibliographic citation.

Au.: Thank you for your observation. The bibliographic citation was added at the end of the sentence.

Rev: Figure 1 the bar indicating the magnification scale contains no micrometer indication. I invite you to refine the photo panel.

Au: The photo panel was refined, and we hope this new version is suitable for evaluation.

Rev: Reading the article carefully, in all the epithelial and mesenchymal tumours studied, even if malignant, I found no reference to potential neoplastic metastases. If your study sample does not include metastatic tumours, I consider it relevant to mention in section 4.1.1 Selection of solid tumour cases.

Au: Thank you for your revision. We included the evaluation of primary lesions and not metastatic tumors. However, some patients had metastatic disease, but we did not collected samples from these lesions. We included this mention in section 4.1.1 for better understanding.

Reviewer 2 Report (New Reviewer)

Comments and Suggestions for Authors

This manuscript describes the protein expression level of six different tyrosine kinase inhibitor (TKI) targets (HER-2, EGFR1, VEGFR-2, PDGFR-b, c-KIT, ERK1/ERK2) in canine solid tumors using immunohistochemistry. Included were five different tumors from 82 different canine patients, namely prostate cancer (not further specified) (n=18), soft tissue sarcomas (n=36), mammary gland tumors (not further specified) (n=20), bladder urothelial carcinomas (n=6) and insulinomas (n=2). Expression profiles were shown to vary between and within different tumor types.

This study investigates relevant, potentially predictive, tumor tissue biomarkers in selected canine cancers. There are some key aspects which have not been covered in the manuscript in its present form. Major revisions are therefore required before the manuscript may be considered for publication.

Major revisions

i)                 For a reliable evaluation and interpretation of the selected tissue biomarkers, the following investigations and comparisons need to be added:

-      Expression benign (e.g. adjacent tissue) vs. neoplastic tissue. A descriptive analysis may be sufficient

-      Expression tumor parenchyma vs. stroma. A descriptive analysis may be sufficient

-        Intratumor expression heterogeneity

ii)                Relevant patient factors, such as age, sex, neutering and breed and their association with the investigated tissue biomarkers need to be added

Minor revisions

-        In the era of digital pathology, semi-quantitative scoring should be replaced by a fully quantitative approach

-        For a better overview, combine all individual tables into one large table

Abstract

-        The types of tumors or tissues investigated would be informative to include

-        Line 21. Rather than different histological subtypes, this refers to different tumor types

-        Line 25. … considered relatively (?) low in the analyzed samples.

Introduction

-        Line 54. … in canine mammary gland tumors and found both HER2… what does ‘both’ refer to?

-        Line 57. … role of lapatinib on HER2 positive and negative mammary gland tumor cell lines.

Results

-        Line 86. … the markers were scored in score… avoid using ‘score’ twice

-        Line 87. Were neoplastic cells clearly distinguishable from stromal and inflammatory cells when analysing the IHC expression (based on histomorphology and the expression pattern)?

-        Line 129. Scattered instead of scatted (check throughout the manuscript)

-        Lines 131-141. There are several inconsistencies between the numbers in the manuscript and in Table 1. Only 16 cases are included in the table.

-        Line 132. Distinguish between glandular, urothelial and mixed prostate carcinoma. Also adapt Table 1 accordingly.

-        Line 148. Mammary gland tumors is too unspecific. Indicated the tumor types.

-        Line 156. Six is a low number of UC. A higher number (minimum 10) would be more convincing

-        Line 162. What was the rationale for including two insulinomas? Only two cases of a tumor entity which is very different from the other investigated cancer types appears random. Either remove this tumor type or add more cases (aiming for at least 10).

Discussion

-        Line 202. … we initially screened different tumor subtypes… What was the tumor selection based on?

-        Line 231. About the expression profile of UC, most of the samples were positive…

-        Lines 232-233. This sentence is difficult to follow. Rephrase.

Material and methods

-        Line 255. … of different tyrosine kinase receptors, formalin-fixed paraffin-embedded tissue blocks from a total of 82 patients were used to perform immunohistochemistry for HER-2, ….

-        Tumor selection criteria are not mentioned and are not clear

-        As mentioned above, the numbers of UC and insulinomas are considered too low

Figure 1

-        As in the manuscript, ‘mammary gland tumor’ is not specific enough. Specify case 3.

Tables

-        No need to add ‘based on the evaluation of the whole tissue fragment’ in the legend.

-        Generate a large table with all investigated tumors instead of the four individual tables

Comments on the Quality of English Language

Only minor English editing is needed.

Author Response

This manuscript describes the protein expression level of six different tyrosine kinase inhibitor (TKI) targets (HER-2, EGFR1, VEGFR-2, PDGFR-b, c-KIT, ERK1/ERK2) in canine solid tumors using immunohistochemistry. Included were five different tumors from 82 different canine patients, namely prostate cancer (not further specified) (n=18), soft tissue sarcomas (n=36), mammary gland tumors (not further specified) (n=20), bladder urothelial carcinomas (n=6) and insulinomas (n=2). Expression profiles were shown to vary between and within different tumor types.

This study investigates relevant, potentially predictive, tumor tissue biomarkers in selected canine cancers. There are some key aspects which have not been covered in the manuscript in its present form. Major revisions are therefore required before the manuscript may be considered for publication.

Major revisions:

Rev: i) For a reliable evaluation and interpretation of the selected tissue biomarkers, the following investigations and comparisons need to be added:

-      Expression benign (e.g. adjacent tissue) vs. neoplastic tissue. A descriptive analysis may be sufficient

Au: Dear reviewer, only in six cases mammary gland tumors, normal surrounding tissue was found. EGFR1, VEGFR-2, PDGFR-β, c-KIT, and MAPK (ERK1/ERK2) were negative in normal tissue from the six cases. However, HER-2 expression was found in normal glands in all six cases. We have included this information in the manuscript.

Rev:  Expression tumor parenchyma vs. stroma. A descriptive analysis may be sufficient

Au: No expression was found the probably because this marker and related to epithelial and cancer cells surface. We have added this information to the results and also discussed this point.

Rev:     Intratumor expression heterogeneity

Au: Dear reviewer, we kindly would like to ask you to be more specific about this point. Intratumor tumor heterogeneity is a very important and complex point. When you mean intratumoral heterogeneity, do you aim for a specific point? If you could explain better, we will be more than happy to add.

Rev:                 Relevant patient factors, such as age, sex, neutering and breed and their association with the investigated tissue biomarkers need to be added

Au: Thank you for your revision. About patient factors, we did not include in this study, since our main goal was to evaluate the expression pattern of solid tumors. In our previous submission, it was proposed to split the paper in two parts (pathological and clinical). We totally agree that this could be important for association but as the journal is focused on molecular purposes we did not include in this first review. We proposed to change the title to PART I, to include only pathological and IHC expressions of the different tumors, and a PART II to correlated with patients factors and treatment. If the reviewer is in accordance with this, we would be glad. If you think it is mandatory this information, then we will describe briefly later.

Minor revisions

Rev:  In the era of digital pathology, semi-quantitative scoring should be replaced by a fully quantitative approach

Au: Dear reviewer, we understand your point and we agree. However, a fully quantitative approach will be poorly applied in a clinical routine since no computational program is approved to perform this analysis in a clinical routine. If the author indicates a program that is applied in clinical routine and could be applied in this study, we will be more than happy to adapt our analysis. In our opinion, the major issue will be the time spent in the analysis for a practical clinical test. In humans, for example, they use semi-quantitative scores for evaluating HER2 analysis and we are trying to follow the same.

- For a better overview, combine all individual tables into one large table

Rev: Thank you for your observations. We merged all tables into one large table as suggested by the reviewer. We hope this new version is suitable for evaluation.

Abstract

Rev:

-        The types of tumors or tissues investigated would be informative to include

-        Line 21. Rather than different histological subtypes, this refers to different tumor types

-        Line 25. … considered relatively (?) low in the analyzed samples.

Au: Thank you for your observation. We included the tumors investigated in line 18, and also modified the sentence suggested.

Rev: Introduction

-        Line 54. … in canine mammary gland tumors and found both HER2… what does ‘both’ refer to?

-        Line 57. … role of lapatinib on HER2 positive and negative mammary gland tumor cell lines.

Au: Thank you for your suggestions and corrections. In line 54 we were referring about HER2 gene amplification and HER2 protein overexpression. We modified the sentence for better understanding in line 55.

Results:

Rev:

-        Line 86 … the markers were scored in score… avoid using ‘score’ twice

-        Line 87. Were neoplastic cells clearly distinguishable from stromal and inflammatory cells when analysing the IHC expression (based on histomorphology and the expression pattern)?

-        Line 129. Scattered instead of scatted (check throughout the manuscript)

-        Lines 131-141. There are several inconsistencies between the numbers in the manuscript and in Table 1. Only 16 cases are included in the table.

-        Line 132. Distinguish between glandular, urothelial and mixed prostate carcinoma. Also adapt Table 1 accordingly.

-        Line 148. Mammary gland tumors is too unspecific. Indicated the tumor types.

-        Line 156. Six is a low number of UC. A higher number (minimum 10) would be more convincing

-        Line 162. What was the rationale for including two insulinomas? Only two cases of a tumor entity which is very different from the other investigated cancer types appears random. Either remove this tumor type or add more cases (aiming for at least 10).

Rev: Thank you for all suggestions. Dear reviewer, yes. Most of the receptors are tyrosine kinase receptors and are expressed in cell membrane. Moreover, they are receptors closely linked to epithelial cells. Therefore, inflammatory and stromal cells are usually negative for these markers. We also tried to include cell round tumors (lymphoma, plasmacytoma and histiocytic tumors) and all tested were negative. We have included this information in results and discussion sections. We reviewed the table and cases and have corrected the numbers of patients in the paragraph about PC (now line 134-144). About mammary gland tumors, we specified in table 1 and in the text (line 152). About insulinoma cases we would like to include solid tumors from endocrine system. We agreed that two cases are too low, so we have increased the tumors from endocrine system to seven. We specified in table 1 and in the text (line 169-176), and we hope this could be acceptable in research for publication.

Rev: Discussion

-        Line 202. … we initially screened different tumor subtypes… What was the tumor selection based on?

-        Line 231. About the expression profile of UC, most of the samples were positive…

-        Lines 232-233. This sentence is difficult to follow. Rephrase.

Au: Thank you for your suggestions. We have modified the sentence in line 209 and in line 242 for better understanding.

Material and methods

Rev:  Line 255. … of different tyrosine kinase receptors, formalin-fixed paraffin-embedded tissue blocks from a total of 82 patients were used to perform immunohistochemistry for HER-2, ….

-        Tumor selection criteria are not mentioned and are not clear

-        As mentioned above, the numbers of UC and insulinomas are considered too low

Au: Thank you for your suggestions and corrections. We agreed that the number of patients is low, and we could not infer totally a fully interpretation, but we would like that the reviewer considered as this is a initial step forward in oncology research. For insulinomas patients, we have changed and include more endocrine tumors considering “endocrine/neuroendocrine system” then we could increase a little the number of samples (N=7). We hope this could be more acceptable.

Rev: Figure 1

- As in the manuscript, ‘mammary gland tumor’ is not specific enough. Specify case 3.

Au: Thank you for your observation. Indeed, is not specific enough, so we specified the subtypes of mammary gland tumors in table 1, we hope this is more suitable for interpretation. Also, we have modified in legend the case 3.

Rev: Tables

-        No need to add ‘based on the evaluation of the whole tissue fragment’ in the legend.

-        Generate a large table with all investigated tumors instead of the four individual tables

Au:  Thank you for your suggestion. We merged all tables into one large table.

This manuscript is a resubmission of an earlier submission. The following is a list of the peer review reports and author responses from that submission.

Round 1

Reviewer 1 Report

Comments and Suggestions for Authors

The present study delves into an intriguing matter that holds broad relevance for the readers of this journal: the screening of various tumor subtypes to ascertain the immunoexpression of distinct tyrosine kinase receptors and its predictive implications for the application of tyrosine kinase inhibitors (TKI) in dogs.

I found the paper remarkably captivating, and it's evident that the authors have invested significant effort in conducting this study. I extend my congratulations to the authors, and without a doubt, I wholeheartedly endorse the publication of this study. I've noted a few suggestions in the attached document.

Wishing you every success.

Warm regards,

Comments on the Quality of English Language

Regarding the English language, I would recommend considering a proofreading service to rectify certain words and expressions. Additionally, it could aid in simplifying certain sentences and paragraphs for enhanced clarity.

Reviewer 2 Report

Comments and Suggestions for Authors

Dear Authors,

I reviewed your manuscript "Immunohistochemistry screening of different tyrosine kinase receptors in canine solid tumors as a potential prediction of therapies", the topic is very interesting and innovative, however the study design has severe critical points. First of all, the introduction does not provide sufficient background on the molecular markers that the authors intend to study in the canine species. The most critical aspect, however, is the division of the study into three "steps", which are not steps of the same study design, but it is an attempt to merge a pathological study with a clinical study. This, in addition to creating a great confusion already visible starting from the abstract and throughout the text, also makes the two aspects - pathological and clinical - poor in details, knowledge and insights. For example, the immunohistochemical evaluation of the various markers analyzed is carried out by counting the number of positive cells and classifying the number in a score, but each marker has a different evaluation from the other, it cannot be generalized in this way. For example, as regards the expression of the kit, it is not so much the number of positive cells that is important as the pattern of expression (membrane, cytoplasmic, perinuclear). Same thing for HER2, there are many studies that identify precise algorithms to judge a sample as positive or negative, it is not enough to count the number of positive cells. Furthermore, no molecular pathology study involving the use of immunohistochemistry can do without pictures. For these reasons, I suggest you to split the study into two articles, one on pathology and the other on clinics, to make the text more understandable and to better investigate the various aspects

Comments on the Quality of English Language

Extensive editing of English language required

Reviewer 3 Report

Comments and Suggestions for Authors

Dear Authors, your paper is original and really interesting but the case recruited are insufficient for supporting the conclusions. Which are  very synthetic due to the lack of scientifically analyzed data according to a statistical approach. It is therefore really necessary to increase the number of observations for each individual group of tumours, if anything by asking for collaboration with veterinary pathologists International group and asking them to make the archives available.